# Dysphagia in Ischaemic Stroke Patients: One Centre Retrospective Study

**DOI:** 10.3390/nu16081196

**Published:** 2024-04-17

**Authors:** Oliwia Maciejewska, Katarzyna Kępczyńska, Małgorzata Polit, Izabela Domitrz

**Affiliations:** 1Bielanski Hospital, 01-809 Warsaw, Poland; maciejewska@autograf.pl; 2Department of Neurology, Faculty of Medicine and Dentistry, Medical University of Warsaw, 01-809 Warszawa, Poland; malgorzata.polit@wum.edu.pl (M.P.); izabela.domitrz@wum.edu.pl (I.D.)

**Keywords:** oropharyngeal dysphagia, ischaemic stroke, hypertension, pneumonia

## Abstract

The aim of this study was to examine the frequency of dysphagia in patients with ischaemic stroke. It was crucial to evaluate the relationship between swallowing disorders and selected demographic and clinical indicators. Additionally, the association between various patient feeding methods and selected demographic and clinical factors was assessed. Based on the analysis of medical documentation, we identified the most important clinical parameters, including demographic data, the frequency of stroke risk factors, the location of the ischaemic lesion, cortical involvement, stroke severity as measured by the NIHSS (Nationale Institutes of Health Stroke Scale), and the methods of feeding post-stroke patients. Dysphagia was observed in 65.9% of the patients in the study group. Hypertension was the most common chronic illness in the studied population of ischemic stroke patients (91.8% of patients). Diet modification (35.7%) and PEG (25%) were the frequent methods of feeding in patients with confirmed dysphagia. Age played a significant role in determining the feeding methods in patients with dysphagia. Patients with a PEG (Percutaneous Endoscopic Gastrostomy) tube were the oldest (79.37 ± 10.80) and 75% of them had pneumonia. Early identification of swallowing difficulties in stroke patients is critical in determining an appropriate and safe feeding plan, as well as initiating logopedics therapy to improve swallowing efficacy and minimize pulmonary complications.

## 1. Introduction

Dysphagia is a disorder that affects the ingestion, fragmentation, and transport of food from the mouth to the stomach. The term is derived from the Greek language, where ‘dys’ means disorder and ‘phagia’ means to eat. In the medical literature, authors may define the problem differently. The European Society for Swallowing Disorders (ESSD) defines dysphagia as ‘difficulty swallowing’. Swallowing disorders are classified as oropharyngeal and oesophageal dysphagias. Neurogenic dysphagia presents during the oral stage of swallowing, causing difficulties in biting, chewing, and oral transport. This type of dysphagia is often associated with facial nerve paresis [1,2].

Swallowing disorders accompany many conditions and their frequency varies depending on etiology, patient age, and information sources (objective assessment or self-assessment). Although dysphagia epidemiology has not been extensively studied, it is estimated that over 5% of the general population experience swallowing disorders [3]. Many sources indicate that the risk of dysphagia increases with age, affecting up to 40% of people over 60 years old [4]. Other studies suggest that the risk may be as high as 73% [5].

Oropharyngeal dysphagia, which is more prevalent than oesophageal dysphagia, affects approximately 13% of the population over 65 years of age and is about as common as diabetes. However, it is most prevalent among geriatric patients with neurological diseases. Studies indicate that the disorder persists in 16% of those aged 70–79 years and increases to 33% for those aged 80 years and older [3].

The epidemiology of swallowing disorders in patients after an ischaemic stroke varies widely. Stroke is the second leading cause of death worldwide, affecting approximately 13.7 million people and resulting in around 5.5 million deaths annually. The incidence of stroke is known to increase with age, with the risk of incident doubling after the age of 55. However, there has been a concerning rise in the incidence of stroke among people aged 20–54 years. This risk has increased from 12.9% to 18.6% between 1990 and 2016 [6,7,8]. The incidence of stroke is influenced by gender. In women, it is higher at a younger age, whereas in men, it slightly increases with age [9]. Dysphagia is a common and highly life-threatening symptom in vascular disorders. Studies have shown incidence rates of dysphagia to range from 19% to 81% [10,11,12,13]. Western literature even suggests 87% [13]. The wide range of percentages can be attributed to the use of various screening and diagnostic techniques, each with different sensitivities. Swallowing disorder is often a downplayed problem, yet it may lead to serious consequences. One of the most common complications of a stroke is the occurrence of respiratory infections. Pneumonia often takes on an aspiration character and results from neurogenic dysphagia of sudden onset [4,14,15,16]. During the acute phase of a stroke, pneumonia can often occur due to neurogenic dysphagia with sudden onset. Recent data suggest that this risk increases threefold. This complication is particularly severe, as it is associated with a high risk of mortality. Studies indicate that dysphagia increases the risk of hospital mortality sixfold and the risk of disability threefold. Dehydration and malnutrition can have serious consequences, including lowered immunity and increased risk of infection, which can lead to reduced physical and cognitive performance. It is important to acknowledge that these complications are scientifically proven. Early detection of swallowing disorders in stroke patients is crucial for adjusting a safe diet, selecting appropriate speech therapy exercises to improve swallowing function, and reducing the risk of pulmonary complications. Dysphagia is a significant diagnostic and care problem, requiring the involvement and cooperation of an entire therapeutic team (physician, nurse, physiotherapist, speech therapist, and psychologist) [4,14,15,16,17]. 

The role of speech therapists is crucial in identifying swallowing disorders. Upon admission, the speech and language therapist conducts screening tests (Gugging Swallowing Screen or Volume-Viscosity Swallow Test) and, in consultation with the doctor, determines the appropriate diet, which may include a normal diet, modifications, Nasogastric Tube, or Percutaneous Endoscopic Gastrostomy (PEG). Various exercises are carried out on the following days to stimulate a return to physiological swallowing. If there is no improvement during this time, the speech therapist, in collaboration with the doctor, performs a Fiberoptic Endoscopic Examination of Swallowing (FEES) to accurately examine swallowing.

The study aimed to analyze the occurrence of oropharyngeal dysphagia in patients with ischaemic stroke, utilizing medical records of patients at the Department of Neurology. We investigated the relationship between swallowing disorders and the selected demographic group, clinical factors (stroke severity, location ischemic changes, and comorbidities such as hypertension, diabetes, dyslipidemia, and heart arrhythmia). The correlations between various patient feeding methods and demographic and clinical methods were analyzed.

The literature contains multiple studies on swallowing disorders in individuals with ischemic stroke. Many of them discuss comparable topics, such as the most prevalent risk factors for dysphagia and the associations of swallowing disorders with specific stroke severity scale scores [4,16]. In contrast, others suggest that there is a correlation between patients with PEG placement and pneumonia [17]. 

## 2. Materials and Methods

### 2.1. General Information

The conducted study was retrospective. The research group consisted of patients from the Stroke Unit of the Department of Neurology at Bielanski Hospital in Warsaw, Poland. The project was conducted with the approval (the approval was granted on 16 May 2022) from the Bioethical Committee of the Medical University of Warsaw, Poland (AKBE/131/2022). 

### 2.2. Study Participants and Sample Size

The study included 350 patients treated after an ischaemic stroke in the Stroke Unit in 2021. The number of patients was calculated using the Sample Size Calculator, which determines the appropriate sample size. A confidence level of 95% was established, with a maximum error of 5%. After performing the calculations, a total of 170 (*N* = 170) patients were obtained. The specified sample size represents a representative sample significant for the planned study.

### 2.3. Clinical Parameters

From the analysis of medical documentation, we have identified the key clinical parameters including demographic data, the frequency of stroke risk factors (such as hypertension, diabetes, dyslipidemia, atrial fibrillation, and previous history of stroke), precise location of the ischaemic lesion, cortical involvement, and severity of stroke as measured by the NIHSS (National Institutes of Health Stroke Scale). Further, patients with swallowing disorders were identified based on the evaluation of the Fiberoscopic Endoscopic Examination of Swallowing (FEES). Possible complications of dysphagia and methods of nutrition in this group were assessed among patients with ischaemic stroke. Methods of feeding post-stroke patients include unmodified independent eating, dietary modification, Percutaneous Endoscopic Gastrostomy (PEG), and Nasogastric Tube feeding.

### 2.4. Statistical Analysis

A random selection process was employed to select the study group from the population of patients with ischemic stroke. IBM SPSS Statistics 28 was used for analysis. The Kolmogorov-Smirnov test was used to test for normal distribution. To investigate the differences in various parameters, parametric tests such as Student’s *t*-test and analysis of variance (ANOVA) as well as non-parametric tests such as the Mann-Whitney *U* test and Kruskal-Wallis test were employed. Additionally, the Bonferroni test (a post-hoc test) was used to determine any significant differences between the groups. The chi-square test of independence was used to compare the groups. Furthermore, the study calculated Cramér’s V coefficient (using multivariate tables) to determine the strength of association between the variables. A significance level of *p* < 0.05 was chosen.

## 3. Results

### 3.1. Population Characteristics

The study involved 170 patients (81 women and 89 men) with ischemic stroke. The mean age was 72.15 years (*SD* = 12.70, range: 19–94). The analysis showed no significant differences between the mean age of women (76.38 years, *SD* = 12.38, range: 19–92) and men (68.29 years, *SD* = 11.79, range: 29–94). Table 1 presents the patient demographics data. 

### 3.2. Frequency of Swallowing Disorders

The analysis showed that 65.9% of the patients had oral-pharyngeal dysphagia, whereas normal swallowing was diagnosed in 34.1% of the patients (Table 2 and Figure 1).

### 3.3. Risk Factors for Ischemic Stroke

This study identified the following risk factors for stroke: hypertension, diabetes, dyslipidemia, cardiac arrhythmia, nicotinism, and history of stroke. In the study group, high blood pressure was the most common concurrent disease with swallowing disorders (92%). Diabetes affected 30% of the patients; dyslipidemia occurred in 35.9% of the patients; rhythm disorders affected 15.3% of the patients; 20.6% declared nicotine addiction; and 13.5% of the patients reported a history of stroke (Table 3).

### 3.4. Methods of Nutrition for Patients with Ischemic Stroke

The study analyzed the dietary regimens of the patients with ischaemic stroke. These regimens included a normal diet, modified texture and consistency of food, Nasogastric Tube, and Percutaneous Endoscopic Gastrostomy (PEG). The patients who had a nasogastric tube longer than 4–6 weeks were planned to PEG. Self-feeding without dietary modifications was the predominant nutrition method among the patients with ischemic stroke, at 48.8%. Subsequently, there were dietary modifications (23.5%), Percutaneous Endoscopic Gastrostomy (PEG) (17.6%), and Nasogastric Tube (10%). The differences were statistically significant with *F* (4.165) = 12.57 and *p* = 0.001 (Table 4).

### 3.5. Analysis of Feeding Methods for Patients with Swallowing Disorders

Modification of diet (35.7%) and Percutaneous Endoscopic Gastrostomy (PEG) (25%) were common methods of feeding patients with confirmed dysphagia. Nasogastric Tube feeding was necessary in 15.2% of the patients with diagnosed dysphagia. The remaining 24.2% did not require any dietary changes. There were statistically significant differences between the variables: *X*^2^ = (1, *N* = 170) = 80.66 and *p* = 0.001. The strength of the relationship between the variables was found to be high (*V Cramér* = 0.79; *p* = 0.001). Table 5 and Figure 2 present the data.

### 3.6. Feeding Methods by Age

Age significantly affected the dietary habits of patients with swallowing disorders. The oldest patients were those with a PEG tube inserted (79.37 ± 10.80). Statistical differences were significant: *F* (4.165) = 4.106 and *p* = 0.003. Patients with a Nasogastric Tube came next in age (75.88 ± 9.13), followed by those on a normal diet (70.26 ± 13.68), and then those on a modified diet (69.68 ± 13.01). Table 6 and Figure 3 present the detailed data.

### 3.7. Prevalence of Pneumonia in Patients with Ischemic Stroke Depending on Feeding Methods

The analysis revealed statistically significant differences in the incidence of pneumonia depending on different feeding methods *X*^2^ = (1, *N* = 170) = 44.92; *p* = 0.001. The risk of pneumonia was significantly elevated in patients with a PEG tube, reaching 75% (*N* = 15). Next, individuals with modified diets experienced a 15% (*N* = 3) incidence rate, followed by those with a Nasogastric Tube at 5% (*N* = 1) and a normal diet at 5% (*N* = 1). Figure 4 presents the data.

### 3.8. Other Analyses

It has been observed that the frequency of swallowing disorders did not have a significant association with gender. The prevalence of dysphagia was similar in both women and men. The analyses using other parameters, including precise localization of the ischemic lesion, hemispheric location, length of hospitalization, and the severity of stroke measured on the NIHSS scale were statistically insignificant.

## 4. Discussion

It was established that 112 patients (66%) of the examined population of 170 study participants suffered from oropharyngeal dysphagia. The diagnosis was based on endoscopic examinations. The analyses correspond to the available literature. A study by Martino R et al. suggests that post-stroke dysphagia affects 37% to 78% of patients. The use of various screening and diagnostic techniques, which vary in sensitivity, accounts for a wide range of percentage values. Additionally, the authors report that the presence of dysphagia during instrumental examination ranges from 64% to 78%. Moreover, they note that the detectability of swallowing disorders when using only screening techniques ranges from 37–45%, while clinical tests yield even higher rates, up to 51–55% [16]. 

The statistics suggest that neurology departments should have access to a flexible nasofiberoscope for an accurate assessment of swallowing disorders, known as Fiberoptic Evaluation of Swallowing Disorders (FEES). The Fiberoptic Evaluation of Swallowing is considered the ‘gold standard’ in the diagnosis of swallowing disorders. The swallowing examination is carried out by a trained team consisting of a doctor, speech therapist, and nurse. The examination should be preceded by a clinical swallowing assessment, which includes taking a detailed history of swallowing problems, performing one of the screening tests, and qualifying the patient for FEES. This prior assessment will allow the selection of appropriate compensatory and adaptive techniques to be tested during the study. Endoscopic examination can be performed at the bedside, even for patients in poor general condition, e.g., on the first day after a stroke [18].

Another screening tool that could be used is a Video Fluoroscopic Swallow Study (VFSS). The Videofluoroscopic Swallow Study (VFSS) is a radiological examination that enables a more precise evaluation of ‘silent aspiration’, which refers to the passage of ingested food into the lower airways without eliciting a cough reflex. The test also assesses the oral phase of swallowing. However, VFSS has limitations, as it cannot evaluate secretions and their potential aspiration, penetration, or retention. The X-ray image only displays the contrast provided by the examiner. The evaluation of aspiration and retention of secretions is one of the primary benefits of the FEES examination [19].

Dysphagia is a common symptom among post-stroke patients, affecting over half of them. It is likely that an independent indicator of worsened health during the recovery period after a stroke. Furthermore, it may cause complications such as malnutrition, dehydration, aspiration, pneumonia, and death [4,14,15]. 

Analyses have shown that men were slightly more among patients with swallowing disorders, with 57 individuals (50.9%) being male, and 55 (49.1%) female. The mean age of the patients was 72.15 years (*SD* = 12.7). Our observations align with the literature reporting that common predictors of dysphagia include age above 60 years and male gender [4,20].

Analyses showed that the most frequent risk factor was arterial hypertension, present in 156 patients (91.8%). Dyslipidemia was the second most common comorbidity, affecting 61 patients (35.9%), followed by diabetes in 51 patients (30%), smoking in 35 patients (20.6%), cardiac arrhythmias in 26 patients (15.3%), and previous stroke in 23 patients (13.5%). The only statistically significant difference was observed in patients with hypertension, with *p* = 0.001. Observations in the present study are consistent with those in other studies [4,21,22]. Arterial hypertension is the main risk factor for stroke, responsible for one-third of the strokes in developed countries and two-thirds in developing countries [4].

The primary objective of treating dysphagia is to reduce the frequency of aspiration and increase oral feeding [23]. In the examined population, the feeding methods of the stroke patients were analyzed. These included: normal diet, where the patient self-feeds, diet modification (related to the consistency and texture of food), Nasogastric Tube feeding, and Percutaneous Endoscopic Gastrostomy (PEG). The applied strategies were highly diverse and tailored individually to the patient’s needs. In the examined population, the predominant feeding method was self-eating without diet modifications, accounting for 83 patients (48.8%). Diet modifications were required by 40 patients (23.5%), whereas 17 (10.0%) needed Nasogastric Tube feeding. Percutaneous Endoscopic Gastrostomy (PEG) was used for 30 patients (17.6%). These differences were statistically significant with *p* = 0.001. An acute vascular incident may cause impaired movement of food along the digestive tract, potentially leading to poor nutrition in patients with swallowing difficulties and hindering their recovery from illness. Nasogastric feeding is a well-established technique that provides nutritional support over a sustained period. However, the application of this treatment should not exceed 2–3 weeks. Prolonged use may result in the development of bedsores on the patient’s mucosa, which can significantly reduce their quality of life. 

Gastrointestinal complications such as diarrhea, constipation, nausea, vomiting, or treatment intolerance may also occur. Mechanical complications include regurgitation, gavage complications, gavage obstruction, or food-borne infections. Metabolic disturbances may also occur. However, it is important to remember that the majority of enteral feeding complications are preventable and are often due to errors in feeding. Another problem is that the patients with dementia may pull the probe out themselves. Care should be taken to ensure that the tip of the probe is well protected and does not irritate the skin. The literature has reported on the importance of nutrition through nasogastric feeding. The insertion of a tube increases the chance of survival by 6%. Subsequently, a Percutaneous Endoscopic Gastrostomy should be considered [20]. Percutaneous Endoscopic Gastrostomy is particularly useful when enteral feeding is prolonged [24]. Furthermore, this method is advantageous as it does not cause any irritation or sores on the nasal cavity’s mucous membrane and does not lead to distension in the upper oesophageal sphincter area [24].

The detailed analysis revealed that among the 112 patients diagnosed with oropharyngeal dysphagia, 27 (24.2%) had a normal diet, 40 (35.7%) had a modified diet, 17 (15.2%) had a Nasogastric Tube inserted, and 28 (25%) required Percutaneous Endoscopic Gastrostomy (PEG) placement. There were statistically significant differences between the variables (*p* = 0.001). These results indicate that the patient’s method of nutrition was adjusted based on the degree of dysphagia severity. It is difficult to determine the criteria for difficulty in swallowing that influenced the selection of the appropriate feeding method. Enteral feeding was probably prescribed for patients with the greatest deficits in swallowing. If there is a low risk of penetration and aspiration, a modified oral diet is given. Modification consists of changing the texture of the food. In future studies, it would be worthwhile to consider which method of feeding is the safest. In a 2015 study, scientists evaluated the effectiveness and safety of using PEG compared to a Nasogastric Tube in adults with swallowing disorders. Both methods did not differ in terms of patient mortality. However, gastrostomy proved to be safer as the Nasogastric Tube feeding was more susceptible to blockages, displacement, dislodgement, or leakage, which can ultimately lead to malnutrition. Researchers also demonstrated that PEG caused fewer gastrointestinal bleeding incidents in comparison to a tube and led to an increase in albumin concentration in the blood [25]. Further analysis, adjusted for age and method of nutrition in patients with ischemic stroke, showed that the greatest differences were observed between nutrition via PEG and normal diet. The age of patients was the highest in the PEG group, with an average of 79 years (*SD* = 10.8). The average age of the patients with a feeding tube was 76 years (*SD* = 9.1), whereas the age for those with modified diets was 70 years (*SD* = 13.1) and also 70 years (*SD* = 13.7) for those with a normal diet. These results show slight deviation from the 2020 study where it was proven that significant differences exist between age and dietary habits. The average age of the patients with a feeding tube was 79 years, whereas for Percutaneous Endoscopic Gastrostomy it was 67 years [24].

An analysis of the incidence of pneumonia based on feeding methods demonstrated statistically significant differences between the parameters (*p* = 0.001). 

The majority of the patients with pneumonia, 15 individuals (75%), had a PEG tube. It is possible that the patients only received a Percutaneous Endoscopic Gastrostomy after being diagnosed with pneumonia. This result indicates the preventive effect of PEG on aspiration pneumonia in stroke patients. Following this group were those with modified diets, 3 patients (15%); followed by those with a Nasogastric Tube, 1 patient (5%); and those with a normal diet, 1 patient (5%).

The results indicate the severity of the issue affecting patients with swallowing disorders. The main purpose of producing PEG is to provide nutrition support for patients with impaired oral feeding abilities. As mentioned earlier, PEG feeding has been shown to be more effective than Nasogastric Tube feeding. It is more comfortable and causes fewer complications. Producing a Percutaneous Endoscopic Gastrostomy (PEG) is considered when the expected period of feeding support exceeds 2–3 weeks. PEG is most commonly indicated for neurological diseases and head and neck cancers. Feeding via PEG typically starts from the day after surgery, although studies have shown that it is equally safe to begin feeding as early as 4 hours post-surgery. The feeding method is determined by the diameter of the gastrostomy tube, the patient’s activity level, and feeding tolerance. If a large-diameter tube is inserted in an active person, they may prepare mixed foods on their own and administer them several times a day in boluses. For patients in a supine position, it may be necessary to administer slow infusions multiple times a day or a slow continuous infusion during sleep to prevent gastroesophageal reflux and regurgitation. The PEG procedure is associated with a low incidence of complications, with severe complications requiring treatment estimated at 1–4%. Complications necessitating surgical intervention occur in only about 0.5% of the cases [26].

The above considerations indicate the seriousness of the problem. The care of the neurological patient should be very careful and adapted to individual needs, disease burden, age, gender, and psychological well-being of the patient.

Therefore, it is beneficial for every neurology department to have a dedicated speech therapist to diagnose dysphagia. In the neurology department, a speech therapist conducts a comprehensive Clinical Swallowing Examination, along with a Fiberoptic Endoscopic Swallowing Examination. Based on the results, the specialist may suggest a suitable diet or dietary plan for the patient. The patient’s diet can be normal, blended, or fragmented depending on their needs. If there is a risk of penetration or aspiration, a nasogastric probe or Percutaneous Endoscopic Gastrostomy may be used. In addition to dietary changes, the speech therapist may recommend exercises to strengthen the muscles involved in the swallowing process, such as those in the tongue, throat, and soft palate. The specialist may select specific maneuvers or techniques, such as bringing the head to the chest or twisting it towards the paresis. Additionally, thermal or gustatory stimulation of the soft palate can increase tongue sensation and induce pharyngeal reflex. Coughing exercises are crucial in preventing fluid aspiration into the trachea [27].

Silent aspiration is particularly dangerous. Silent aspiration occurs without triggering the cough reflex. The important thing about silent aspiration is that there are two aspects to it. The first is the aspiration of saliva and the second is the aspiration of swallowed fluid or food. The literature reports that a reduced cough reflex is caused by aging and cerebrovascular disease, especially in the acute stage [28,29,30]. An impaired cough reflex can lead to pneumonia [31]. 

A diminished or absent cough reflex is caused by a decreased sensation of the mucosa innervated by the superior laryngeal nerve. It is therefore very difficult to diagnose based on clinical examination alone. Instrumental studies, i.e., Fiberoptic Endoscopic Evaluation of Swallowing Disorders and Videofluoroscopy Swallowing Study, are essential. However, there are situations where it is not possible to perform these tests. There are reports in the literature that use citric acid or tartaric acid nebulization to assess cough function. Studies show that this test is rarely used to screen for silent aspiration. The above methods can sometimes be difficult for the patient, invasive (e.g., suctioning) or ineffective [27].

Research suggests that nebulization with tartaric acid was effective in inducing cough in patients with dysphagia, particularly in those with silent aspiration. This method may be useful for inducing cough in patients with silent aspiration. Other studies have reported that capsaicin may also act as a cough stimulant [32,33]. Studies have been found to use citric acid and tartaric acid to induce cough and increase expectoration efficiency. Previous and typical methods based on respiratory protection and centered on coughing are direct instructions to the patient to cough. Another method is rehabilitation based on phonation and breathing exercises. The third method is endotracheal suctioning. However, patients with dysphagia may have difficulty coughing voluntarily, especially those who have had a stroke [34,35].

Another important point to note is the prevalence of post-stroke depression. The authors describe post-stroke depression as a disorder characterized by persistent emotional decline and reduced interest. Studies show that post-stroke depression occurs mainly within the first year after stroke. Sources indicate that it is most common within the first three months after stroke, but some studies have shown that the incidence of poststroke depression is the highest one year after stroke. Depression is very serious because it reduces the motivation to rehabilitate and thus reduces the reversal of the effects of stroke. It can also lead to suicide [36].

Dysphagia is a very serious problem for people with vascular disease, as shown in the above discussion. In future studies, it would be worthwhile to focus on comparing the presence of pneumonia and the results from the NIHHS Scale. In the studies from Ohio, scientists demonstrated that the aforementioned two parameters (NIHSS scale and pneumonia presence) are the best predictors of the need to perform PEG insertion in patients with ischaemic stroke. As a result, benefits such as early decision-making regarding appropriate nutrition, shorter hospitalization, and potential cost savings will be present. The analysis revealed that patients with pneumonia and NIHSS score ≥ 12 had the greatest likelihood of undergoing PEG insertion [17]. Future research on cough induction therapy should consider the frequency, timing, and safety of acid nebulization in cough induction therapy. Research into the relationship between swallowing problems and the presence of symptoms of post-stroke depression would also be worthwhile.

## 5. Conclusions

The findings suggest a high prevalence of dysphagia in individuals following an ischaemic stroke. Early identification of swallowing difficulties in stroke patients is critical in determining an appropriate and safe feeding plan and initiating speech therapy to improve swallowing efficacy and minimize pulmonary complications. Consequently, dysphagia presents a significant diagnostic and patient care issue that necessitates the collaboration of multiple healthcare providers such as doctors, nurses, physiotherapists, speech therapists, and psychologists.

## Figures and Tables

**Figure 1 nutrients-16-01196-f001:**
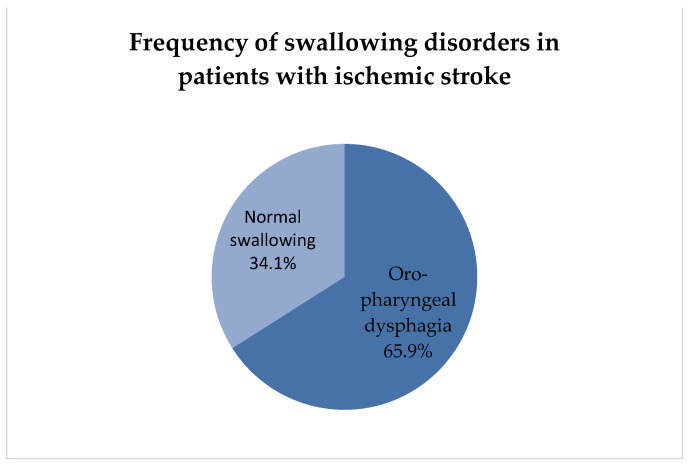
Frequency of swallowing disorders in patients with ischemic stroke.

**Figure 2 nutrients-16-01196-f002:**
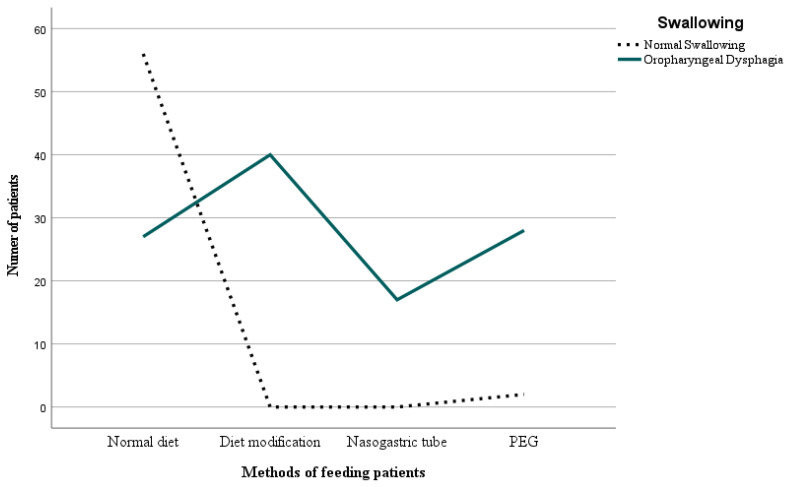
Analysis of feeding methods for patients with swallowing disorders. PEG: Percutaneous Endoscopic Gastrostomy.

**Figure 3 nutrients-16-01196-f003:**
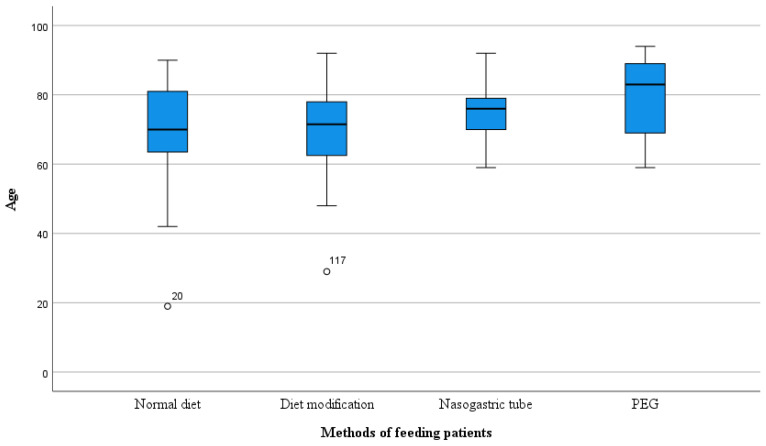
Feeding methods by age. PEG: Percutaneous Endoscopic Gastrostomy.

**Figure 4 nutrients-16-01196-f004:**
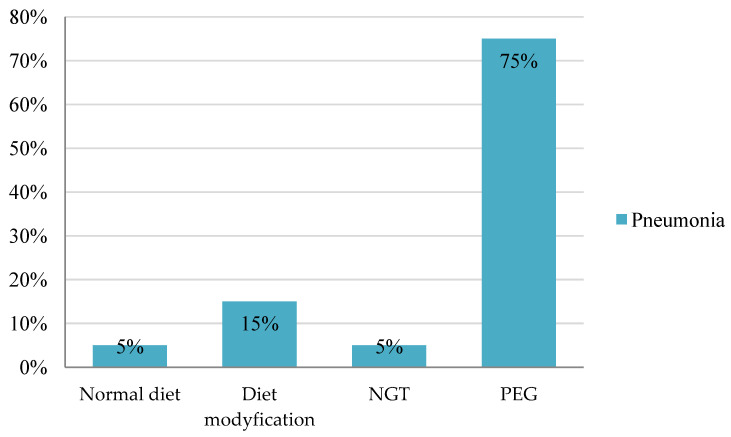
Prevalence of pneumonia in patients with ischemic stroke depending on feeding methods. NGT: Nasogastric Tube, PEG: Percutaneous Endoscopic Gastrostomy.

**Table 1 nutrients-16-01196-t001:** Demographic data for patients with ischemic stroke.

	*N*	[%]	Mean Age [Age ± *SD*]
Total patients	170	100	72.15 ± 12.70
Women	81	47.6	76.38 ± 12.38
Men	89	52.4	68.29 ± 11.79

*SD*: standard deviation, *N*: number of patients.

**Table 2 nutrients-16-01196-t002:** Frequency of swallowing disorders in patients with ischemic stroke.

Name	*N*	[%]
Normal swallowing	58	34.1
Oropharyngeal dysphagia	112	65.9

*N*: number of patients.

**Table 3 nutrients-16-01196-t003:** Frequency of incidence of risk factors for ischemic stroke in the examined patient population.

Risk Factor	*N*	[%]
Hypertension	156	91.8
Diabetes	51	30
Dyslipidemia	61	35.9
Arrhythmia	26	15.3
Nicotine dependence	35	20.6
Prior history of stroke	23	13.5

*N*: number of patients.

**Table 4 nutrients-16-01196-t004:** Methods of nutrition for patients with ischemic stroke.

Name	*N*	[%]	*p*-Value
			0.001 ^a^
Normal diet	83	48.8	
Diet modification	40	23.5	
Nasogastric tube	17	10.0	
PEG	30	17.6	

*N*: number of patients, PEG: Percutaneous Endoscopic Gastrostomy. ^a^ Univariate analysis of variance.

**Table 5 nutrients-16-01196-t005:** Analysis of feeding methods for patients with swallowing disorders.

	Normal Swallowing		Oropharyngeal Dysphagia		Total		*p*-Value
	*N*	%	*N*	%	*N*	%	
							0.001 ^a^
Normal diet	56	96.7	27	24.2	83	48.8	
Diet modification	0	0.0	40	35.7	40	23.5	
NGT	0	0.0	17	15.2	17	10.0	
PEG	2	3.4	28	25.0	30	17.6	
Total	58	100.0	112	100.0	170	100.0	

*N*: number of patients, NGT: Nasogastric Tube, PEG: Percutaneous Endoscopic Gastrostomy. ^a^ Pearson’s Chi-square test.

**Table 6 nutrients-16-01196-t006:** Feeding methods by age.

	Normal Diet	Diet Modification	NGT	PEG	*p*-Value
Patients (age ± *SD*)	70.26 ± 13.68	69.68 ± 13.01	75.88 ± 9.13	79.37 ± 10.80	0.003 ^a^

*SD*: standard deviation, NGT: Nasogastric Tube, PEG: Percutaneous Endoscopic Gastrostomy. ^a^ Univariate analysis of variance.

## Data Availability

Any further requested information regarding the experimental and data analysis during the current study is available from the corresponding author on request. It is possible that other researchers may not be able to comprehend the data I have collected, or may use it for purposes that are not aligned with my original intentions.

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
