# Peer review of "Dysphagia in Ischaemic Stroke Patients: One Centre Retrospective Study"

_nutrients, 2024, doi:10.3390/nu16081196_

Round 1
Reviewer 1 Report
Comments and Suggestions for Authors
Dear authors, I’ve read your interesting article,
Under Study Participants you write, the following
“The study included patients treated after ischemic stroke in the Stroke Unit in 2021 was 350 patients. Based on this, a group of 170 patients was identified”
Please explain how those patients were selected, and why the others were excluded.
In the discussion you write
The good news is that in over half of the surveyed individuals, the diet and method of administering food were modified.
What do you mean by that?
You also write
However, gastrostomy proved to be safer as the tube was more susceptible to blockages, displacement, dislodgement or leakage, which can ultimately lead to malnutrition.
Should that not be, is less susceptible because a PEG tube is much less susceptible to all that sort of things than a nasogastric one
You also write
An analysis of the incidence of pneumonia based on feeding methods demonstrated statistically significant differences between the parameters (p=0.001). The majority of patients with pneumonia had a PEG tube
Are you not turning cause and consequence around because there is no reason why that sort of tube would lead to pneumonia, as a matter-of-fact it protects the patient from getting one.
You also write
Early identification of swallowing difficulties in stroke patients is critical in determining an appropriate and safe feeding plan, initiating speech therapy to improve swallowing efficacy and minimizing pulmonary complications.
That seems logical, but there’s nothing about that in your article so please elaborate on that one in your discussion for example.
Please add something in your article about how many patients who had a nasogastric tube had to be changed to a PEG one and why that was needed.
Quality of English language in general is okay, but there are a number of sentences that are grammatically incorrect or are unclear.
Author Response
Dear Reviewer,
Thank you for reviewing our manuscript entitled " Dysphagia in Ischaemic Stroke Patients: One Centre Retrospective Study.”.
The comments are very helpful and this has certainly helped to further improve our contribution. All changes have been highlighted using track-changes in the revised document.
We hope that our revised manuscript can now be considered for publication in the Nutrients.
Based the comments and suggestions we have made changes, which are detailed below in the attachment.
Kind regards,
Katarzyna Kępczyńska,
The corresponding author

Reviewer 2 Report
Comments and Suggestions for Authors
Dear authors,
Your article untitled « Dysphagia in Ischaemic Stroke Patients: One Centre Case-control Study » focus on an important but few studied topics. Your study aims at examining the frequency of dysphagia in patients with ischaemic stroke. Your results permitted to conclude that early identification of swallowing difficulties in stroke patients is critical in determining an appropriate and safe feeding plan, as well as initiating logopedics therapy to improve swallowing efficacy and minimizing pulmonary complications. However, you should consider some comments.
Title
1. Your study is an observational study and not a case control study. Please modify your tittle.
Introduction
1. “The project aimed to analyze the occurrence of oropharyngeal dysphagia… as well as the duration of hospital stay.” This paragraph should be revised. It is a mix between the goal of the study and a summary of the analysis of the study. You should only give the aim of the study.
2. “The analyses conducted in the present study are novel as they have not previously been explored before.” I am not sure that this sentence is important. In addition, it is surprising because in the previous paragraph, you explained that several articles discussed “comparable topics”. Thus, we don’t understand the novelty of your study
Material and methods
1. Are you sure that your study is a case control study? For me, it is an observational study and you have to report you data according to STROBE guidelines and add the checklist in supplementary files.
2. Why have studied 170 patients if you had 350 patients available?
3. “The number of patients was calculated using the Sample Size Calculator, which determines the appropriate sample size. A confidence level of 95% was establised, with a maximum error of 5%.” First, you have to correct “established”. Second, you have to put this in a section “sample size” and to add the number obtained.
4. Add the date of the study
5. Please add the main outcomes and the secondary outcomes
Results
1. Table 1 Did you have other information? marital status? Active or retired? …
2. “Self-feeding without diet ary” Please correct “diet ary”
Please check all your manuscript because there is a lot of double spaces, spelling errors…
Author Response

(The authors gave the same response as above.)

Reviewer 3 Report
Comments and Suggestions for Authors
Dysphagic issues have an impact on patients' health, clinical outcomes, and the strain that contemporary health care systems are under due to direct and indirect expenditures as well as society. The authors discussed a case-control study conducted at a single center on dysphagia in ischemic stroke patients. The study included 170 participants who had suffered an ischemic stroke.
Please indicate if this is an observational or retrospective study in the methodology section.
Name the IBM SPSS Statistics version that was used for the analysis.
I propose referencing the correlation among malabsorption, malnourishment, and dysphagia (10.1136/bcr.07.2008.0358; 10.1097/MCG.0b013e318159c654); specifically in neurological disorders (10.1016/j.ncl.2009.09.007).
Author Response

(The authors gave the same response as above.)

Round 2
Reviewer 1 Report
Comments and Suggestions for Authors
Thank you for making the changes..
Author Response
Dear Reviewer,
Thank you for reviewing our manuscript entitled " Dysphagia in Ischaemic Stroke Patients: One Centre Retrospective Study”.
Kind regards,
Katarzyna Kępczyńska,
The corresponding author
Reviewer 2 Report
Comments and Suggestions for Authors
Dear authors,
thank you for this new version that dress some of my comments but could you answer to the remaining comments:
Material and methods
1. you Can you report your data according to STROBE guidelines and add the checklist in supplementary files.
2. Why have studied 170 patients if you had 350 patients available?
4. Add the date of the study
5. Please add the main outcomes and the secondary outcomes
Results
1. Table 1 Did you have other information? marital status? Active or retired? …
Author Response
Dear Reviewer,
Thank you for reviewing our manuscript entitled " Dysphagia in Ischaemic Stroke Patients: One Centre Retrospective Study.”.
The new comments are very helpful and this has certainly helped to further improve our contribution. All changes have been highlighted using track-changes in the revised document.
Kind regards,
Katarzyna Kępczyńska,
The corresponding author
Response of the authors to the Reviewer’s Report
Manuscript: “Dysphagia in Ischaemic Stroke Patients: One Centre Retrospective Study.”
Authors:
Oliwia Maciejewska2, Katarzyna Kępczyńska1, Małgorzata Polit1, Izabela Domitrz1
1Department of Neurology, Faculty of Medicine and Dentistry, Medical University of Warsaw, Bielanski Hospital, Poland
2 Bielanski Hospital, Warsaw, Poland
Based the comments and suggestions we have made changes, which are detailed below.
Reviewer 2
- you Can you report your data according to STROBE guidelines and add the checklist in supplementary files.
Answer:
We reported the data according to STROBE guidelines. The checklist has been attached.
- Why have studied 170 patients if you had 350 patients available?
Answer:
The sample size for the study was calculated using a Sample Size Calculator.
Using this calculator, a representative number of patients was determined as follows:
The study included patients treated after ischemic stroke in the Stroke Unit in 2021 was 350 patients. The number of patients was calculated using the Sample Size Calculator, which determines the appropriate sample size. A confidence level of 95% was established, with a maximum error of 5%. After performing the calculations, a total of 170 (N = 170) patients were obtained. The specified sample size represents a representative sample, significant for the planned study.
- Add the date of the study
Answer:
The date of the the study and of the approval from Bioethical Committee of the University of Warsaw was added as follows.
The documentation was analyzed in the period from June to December 2022 after the approval the consent from the Bioethical Committee. The project was conducted with the approval (the approval was granted on 16th May 2022) from the Bioethical Committee of the Medical University of Warsaw(Poland) (AKBE/131/2022).
- Please add the main outcomes and the secondary outcomes
Answer:
We added those informations as follows:
3.7. Prevalence of pneumonia in patients with ischemic stroke depending on feeding methods
The analysis revealed statistically significant differences in the incidence of pneumonia depending on different feeding methods X2=(1, N=170)=44.92; p=0.001. The risk of pneumonia was significantly elevated in patients with a PEG tube, reaching 75% (N = 15). Next, individuals with modified diets experienced a 15% (N = 3) incidence rate, followed by those with a naso-gastric tube 5% (N = 1) and a normal diet 5% (N = 1).
Results
- Table 1 Did you have other information? marital status? Active or retired? …
Answer:
We didn’t check those informations.
